# Bergamottin and PAP-1 Induced ACE2 Degradation to Alleviate Infection of SARS-CoV-2

**DOI:** 10.3390/ijms232012565

**Published:** 2022-10-19

**Authors:** Mengjia Li, Yongzheng Zhang, Amir Zeb, Yang Wu, Lufeng Cheng

**Affiliations:** Department of Pharmacology, School of Pharmacy, Xinjiang Medical University, Urumqi 830011, China

**Keywords:** ACE2, RBD Spike S, PAP-1, bergamottin, COVID-19

## Abstract

Angiotensin-converting enzyme 2 (ACE2), a functional receptor for SARS-CoV, now appears likely to mediate 2019-nCoV entry into human cells. However, inhibitors such as PAP-1 and bergamottin have been discovered; both of them can preferentially bind to ACE2, prevent RBD Spike S protein from binding to ACE2, and reduce the binding sites for RBD Spike S protein. In addition, we investigated the binding energy of PAP-1 and bergamottin with ACE2 through molecular docking with bio-layer interferometry (BLI) and found relatively high binding affinity (KD = 48.5 nM, 53.1 nM) between the PAP-1 and bergamottin groups. In addition, the nanomolar fraction had no effect on growth of the AT-II cell, but 150 µM PAP-1 and 75 µM bergamottin inhibited the proliferation of AT-II cells in vitro by 75% and 68%, respectively. Meanwhile, they significantly reduced ACE2 mRNA and proteins by 67%, 58% and 55%, 41%, respectively. These results indicate that psoralen compounds PAP-1 and bergamottin binding to ACE2 protein could be further developed in the fight against COVID-19 infection during the current pandemic. However, attention should be paid to the damage to human alveolar type II epithelial cells.

## 1. Introduction

Since December 2019, a novel coronavirus (SARS-CoV-2) causing severe acute respiratory syndrome, also known as COVID-19, has spread rapidly around the world [1]. As of August 2022, 596,119,505 confirmed cases of COVID-19 have been reported globally, including 6,457,101 deaths, according to the World Health Organization (WHO). So far, five new coronavirus variants have been identified as “variants of concern” by the WHO: Alpha (B.1.1.7), Beta (B.1.351), Gamma (P.1), Delta (B.1.617.2), and the latest, Omicron (B.1.1.529). The emerging variants and mutations of SARS-CoV-2 have posed a great challenge to public health and safety.

COVID-19 is characterized by fever, dry cough, and fatigue. Severe cases rapidly progress to acute respiratory distress syndrome (ARDS), septic shock, refractory metabolic acidosis, coagulation dysfunction, and multiple organ failure.

A research study has shown that the SARS-CoV-2 spike protein (S protein) can directly bind with highly enriched ACE2 (angiotensin-converting enzyme 2) in type II alveolar epithelial cells with high affinity [2], and ACE2 becomes a key target for entry into the body of the novel coronavirus [3]. ACE2 mainly degrades Ang-II to produce Ang (1–7). ACE2 plays a key role in maintaining the balance between the vasoconstrictor axis of the AC—Ang-II—AT1 receptor and the vasodilation axis of the ACE2—Ang(1–7)—MAS receptor [4,5].

COVID-19 releases viral genetic material via ACE2 to invade the human body. During the early period, patients show progressive decline in peripheral blood lymphocytes, progressive increase in inflammatory factors and C-reactive protein, and rapid progression of lung lesions into severe and critical clinical warning indicators within a short period of time. In the middle period, a large number of cytokines are generated to form a “waterfall effect” which intensifies the immune response and leads to severe lung inflammation with a large number of inflammatory substances permeating and blocking the airway, ARDS alveolar microthrombus formation, or even death. This is mainly due to the imbalance of the immune regulation network, which leads to the accumulation of a large number of immune cells (such as Th1 and Th2) and immune factors (IL-6, TNF-α, and TGF-β), and progresses to tissue and organ damage [6,7].

Research has found that more than 30 medicines, including western medicines, natural products, and traditional Chinese medicine, may have potential efficacy against COVID-19. Some of these medicines and traditional Chinese medicine compounds have been verified in clinical research [8].

A new approach is currently being developed for the treatment of patients with COVID-19. To date, the US FDA has approved the monoclonal antibodies molnupinavir and remdesivir, and EUA has approved Paxlovid (nirmatrelvir plus ritonavir) for treating the COVID-19 infection [9]. These drugs mainly target the viral replication cycle, virus entry and translocation to the nucleus. Some drugs can also enhance the innate anti-virus immune response. Furthermore, many drugs can treat COVID-19 by inhibiting the virus-mediated cytokine storm [10,11,12]. Remdesivir has been shown to be a potential therapeutic agent that effectively shortens the recovery time and reduces respiratory tract infections in hospitalized adult patients with COVID-19. The results of recent clinical trials of chloroquine have provided new information on its use in patients with COVID-19, but adjuvant therapy in several patients with acute SARS-CoV-2 infection has shown that higher doses of chloroquine are lethal.

Under the existing technical conditions, the feasible prevention and treatment methods for COVID-19 include vaccines, small molecule drugs, traditional Chinese medicine, antibodies, etc. [13]. At present, the main treatment methods in China focus on traditional Chinese medicine concerned with regulating the body state and improving immunity [14,15]. This reflects the importance of traditional Chinese medicine in the fight against COVID-19 in China [16].

Psoralen belongs to the furanocoumarin compounds which are commonly used in the treatment of skin diseases such as vitiligo and psoriasis. In addition, they also have anti-tumor, antibacterial, antiviral, bronchodilator, and anticoagulant effects. Psoralen is distributed in a variety of plants such as Psoralen, Fructus Citri Limoniae, Radix Glehniae, and Rhizoma Et Radix Notopterygii [17]. Its characteristics include strong pharmacological activity, low toxicity, high bioavailability, and good curative effect [18].

According to reports [19] and using molecular docking software, coumarin compounds such as psoralen, bergapten, heraclenin, and angelicin can significantly inhibit one of the major proteases Mpro (also known as 3CLpro, PDB ID: 5N5O) of COVID-19. Mpro can cleave 12 smaller proteins out of a polyprotein after the viral RNA enters human cells and participates in the replication of viral RNA [20], thus predicting the potential anti-COVID-19 effects of the above compounds. Although vaccination is the critical solution for COVID-19 prevention and pandemic control [21], the epidemic situation is very different between regions. Furthermore, the virus is still mutating. Even though antiviral drugs are being used against COVID-19, there is still an urgent need for additional safe and effective drugs to combat SARS-CoV-2. Therefore, we explored the antiviral effect of compounds PAP-1 and bergamottin, which is the ACE2 protein interaction, against SARS-CoV-2 infection in vitro during the current pandemic.

## 2. Results

### 2.1. ACE2 Direct Protein Target of PAP-1 and Bergamottin

To lock the interaction between the psoralen compounds and ACE2 protein, we used the drug-likeness module to screen out the psoralen compounds followed by screening with the absorption, distribution, metabolism, and excretion/toxicity (ADME/T) module to identify the top three analogue molecules from the TCMSP library with acceptable ADME/T properties (Table 1). To investigate the interaction of ACE2 with PAP-1 and bergamottin, specific binding between the two compounds was examined via docking tools to predict the binding of the analogue molecules with ACE2. We found that PAP-1 and bergamottin were the preferred molecule based on the binding energy, drug-likeness properties, and binding energies (Figure 1A). A pose of PAP-1 and bergamottin with ACE2 are shown in Figure 1B.

Next, we further investigated the PAP-1 and bergamottin interactions with ACE2 using a BLI assay. The following two psoralen compounds concentration were selected: 3.13, 6.25, 12.5, 25, 50, and 100 µM. To improve the accuracy of the binding experiment, the tests by the Octet system were carried out in duplicate. The experiment was performed in triplicate, as shown in Figure 2 and Table 2 and Table 3. Importantly, the RBD Spike S protein with ACE2 protein binding energy was 7.02 × 10^−10^ M (Figure 2A). The BLI assay confirmed that there was direct interaction between two psoralen compounds and ACE2 and the RBD Spike S proteins, and the Kd value of bergamottin binding to ACE2 was 53.1 µM (Figure 2B and Table 2) while the PAP-1 binding to ACE2 was 48.5 µM (Figure 2D and Table 2). Furthermore, the Kd value of bergamottin binding to the RBD Spike S proteins was 122 µM (Figure 2C and Table 3). There was similar direct interaction between PAP-1 and the RBD Spike S proteins (Figure 2E and Table 2), and there was significant direct interaction as compared to bergamottin with the RBD Spike S proteins. Collectively, these results indicate that PAP-1 and bergamottins directly bind to ACE2.

### 2.2. Molecular Dynamic Model Simulation of PAP-1 and Bergamottin Binding to ACE2 Protein

The glide method was used for docking and we found that bergamottin could bind to ACE2 better than PAP-1, but the binding ability was general. However, PAP-1 was found to have a better binding capacity than bergamottin by the induced fit-docking method, as shown in Table 4.

A molecular dynamics analysis showed that PAP-1 could bind to the cavity of ACE2 (Figure 3A). PAP-1 could bind to six residues in the cavity (Figure 3E), followed by ASN210, LEU95, LYS562, and PRO565 (Figure 3C and Appendix A). Further analysis showed that the four atoms (6, 9, 14, and 25) of PAP-1 could cause conformational changes of the ACE2 protein (Figure 3D). It is noteworthy that 4-phenoxybutoxy is the key for PAP-1 to exert its effects. In terms of drug action, bergamottin can form hydrogen bonds with ASN residues of ACE2 protein (Figure 3B and Appendix A). The analysis of binding stability showed that the conformational change of the drug was small (RMSD < 3), as shown in Figure 3F.

Furthermore, using a molecular dynamics simulation, we found that bergamottin could exist in the cavity where the active site of the ACE2 protein was located, as shown in Figure 4A. A RMSF analysis of the ACE2 protein showed that the conformation of the active site of the protein changed significantly due to the fact that bergamottin could bind to eight residues in this region (Figure 4E) followed by ASP350, ARG393, PHE390, and PHE40 residues (Figure 4C). Further analysis of the molecular structure revealed that the eight atoms (4, 7, 9, 11, 13, 19, 21, and 24) of bergamottin could cause conformational changes of the ACE2 protein (Figure 4D and Appendix A). It should be noted that the 3,7-dimethylocta-2,6-dienoxy group of the drug was the key to affecting the protein conformation. Next, we analyzed the drug action force and found that the protein binding of bergamottin and ACE2 was mainly water bridge and hydrophobic bonds (Figure 4B). However, we found that the conformational change of the drug was relatively large (RMSD > 3) when we analyzed the RMSD of the protein and ligand, suggesting that the binding stability was relatively low.

### 2.3. PAP-1 and Bergamottin Inhibit the Proliferation of AT-II Cells In Vitro

In the control group, alveolar type II cells were epithelioid with polygonal morphology and cultured in the form of a monolayer. The adjacent cells were connected by tight junction or intermediate junction, and cells were serum-free starved for 8 h. The cell morphology was unchanged. After being co-cultured in complete medium for another 48 h, the cells in the control group proliferated significantly. Compared with the control group, in the PAP-1 group the cell morphology changed from polygonal to long spindle-shaped, and the tight junction between the adjacent cells was damaged significantly (Figure 5A). Meanwhile, in the bergamottin group only the cell morphology changed, but the tight junction between adjacent cells was intact and no obvious damage was noted (Figure 6A).

The CCK-8 results revealed AT-II cell activity. Compared with the control cells, the results showed that PAP-1 with AT-II resulted in a significant decrease of cell viability at 24, 48, and 72 h; however, the nanomolar fraction had no effect on growth of the AT-II cells (Figure 7A–C). Micromolar grade PAP-1 significantly reduced cell viability at 24, 48, and 72 h, but compared with 24 or 72 h, the 48 h and 150 µM PAP-1 significantly inhibited cell viability (Figure 5B). Clearly, PAP-1 inhibited the AT-II cells proliferation. After 48 h of treatment of AT-II cells with PAP-1 at 150, 200, 300, 400, and 800 µM (*n* = 5) (Figure 5C), cell viability of the AT-II cells was inhibited in a concentration-dependent manner in accordance with the Hill equation: y = ax^b^/(c^b^ + x^b^). The IC50 value was 153.44 µM. At a concentration of 400 µM or 800 µM, PAP-1 suppressed cell viability at the maximum inhibition rate of 80% in AT-II cells, after which the viabilities stabilized. Accordingly, 48 h and 150 µM was chosen as the PAP-1 dose for subsequent experiments.

Compared with the control cells, the results revealed that bergamottin with AT-II resulted in a significant decrease of cell viability at 24, 48, and 72 h. The nanomolar fraction had no effect on growth of AT-II cells (Figure 7D–F). Bergamottin significantly reduced cell viability at 24, 48, and 72 h (Figure 6B), and 25 µM bergamottin inhibition rate at 24, 48, and 72 h was 27.1%, 2.81%, and 2.61%, respectively. After 24 h of treatment of AT-II cells with bergamottin at 25, 50, 75, 100, and 150 µM (*n* = 5) (Figure 6C), the cell viability of the AT-II cells was inhibited in a concentration-dependent manner in accordance with the Hill equation: y = ax^b^/(c^b^ + x^b^). The IC50 value was 68.49 µM. At a concentration of 100 µM or 150 µM, bergamottin suppressed cell viability at the maximum inhibition rate of 70% in AT-II cells, after which the viabilities stabilized. Accordingly, 24 h and 70 µM was chosen for bergamottin as the dose for subsequent experiments.

### 2.4. ACE2 Is Expressed in Human AT-II Cells

Several studies have demonstrated ACE2 expression in the epithelial cells of tissues including the lungs. In order to determine whether ACE2 is also expressed in AT-II cells, we performed a Western blot analysis of AT-II cells obtained from ACE2 proteins. Western blot analysis clearly showed that the ACE2 protein was expressed in AT-II cell lysates as a single 110 kD band. PAP-1 and bergamottin significantly reduced ACE2 proteins by 55% and 41%, respectively (Figure 5D and Figure 6D).

### 2.5. Quantitative PCR with Reverse Transcription Analysis—ACE2 Is Expressed in Human AT-II Cells

Meanwhile, we performed qPCR analysis of the AT-II cells obtained from ACE2 mRNA. The qPCR analysis clearly showed that ACE2 mRNA was expressed in AT-II cells. PAP-1 and bergamottin significantly reduced ACE2 mRNA by 67% and 58%, respectively, as was confirmed by the expression of ACE2 molecules in AT-II cells. PAP-1 and bergamottin significantly reduced ACE2 proteins and mRNA, as was confirmed by the expression of ACE2 molecules in AT-II cells with BLI (Figure 5E and Figure 6E).

## 3. Discussion

COVID-19 invades the human body and releases its genetic material through ACE2 to stimulate the body’s immunity, causing the secretion of a large number of cytokines (immune storm) and depositing micro-thromboses to aggravate injury to the lungs, heart, kidney, and other important organs. Researchers are now working to tackle this global problem [22]. At present, an important measure of the ongoing efforts to control and mitigate the impact of the COVID-19 pandemic has been to reduce the transmission of the virus between individuals [23]. The purpose of this project is to explore the potential active compounds for the prevention and treatment of COVID-19. In our study, we found human alveolar type-II epithelial cells showed high expression of ACE2 proteins across all human alveoli [24].

Here, we report on PAP-1 and bergamottin, a natural small molecule, which are basically psoralen compounds. Psoralen compounds were docked with ACE2 and RBD the Spike S protein (PDB ID: 7DF4), and the preliminary prediction of comprehensive binding energy showed that bergamottin and PAP-1 had high binding ability with the ACE2 protein. Molecular docking tools showed that the two compounds formed a stable hydrogen bond, and the structure was more stable and more easily entered into the 3D structural cavity of ACE2. The five-position electron cloud had lower density, lower molecular energy, and higher stability, and was more likely to bind with the RBD Spike S protein and ACE2 protein. The higher density of the eight-position electron cloud made it difficult to bind to the two proteins. However, Autodock gives scores with negative values. Over time, different search algorithms have been developed for rigid body docking and flexible docking. We utilized the genetic algorithm while searching for the best rigid pose in blind docking using the AutoDock tools, and the force field-based scoring system was used for interpretation of the docking poses both in the AutoDock tools and SYBYL. While in the discovery studio, we utilized the systemic conformational search algorithm. The ligand was allowed to dock freely, and the default scoring function LigScore was used. Due to different algorithms and scoring functions, the results displayed some differences, but the RMSD score for overall docking remained stable and fell within the range of 2 Å.

After mixing the RBD Spike S protein with the ACE2 protein, the Octet RED96e molecular-protein interaction instrument was used to record and fit the binding curves of different concentration gradients and to obtain the binding energy parameter KD and other parameters. We found the RBD Spike S protein with ACE2 protein binding energy was 7.02 × 10^−10^ M, which is consistent with reports in the literature [25]. Next, we sought to further investigate bergamottin or PAP-1 and ACE2 and RBD Spike S protein interactions using a BLI assay. Bergamottin and PAP-1 binding with ACE2 protein resulted in high energy, 5.31 × 10^−^^5^ M and 4.85 × 10^−5^ M, respectively. Bergamottin and PAP-1 binding with the RBD Spike S protein resulted in energy of 4.86 × 10^−^^5^ M and 1.22 × 10^−^^4^ M, respectively. Bergamottin binding with the RBD Spike S protein was better than PAP-1, but bergamottin can preferentially bind to ACE2. However, mixing the RBD Spike S protein with the ACE2 protein after adding it to PAP-1 and bergamottin showed that there was no typical binding kinetics between the two proteins, and the curves could not be fitted. Therefore, KD and other relevant parameters could not be obtained, indicating that the protein was obviously denatured and precipitated. In order to analyze this phenomenon, we used the Glide module of Schrodinger software to blind dock the two compounds with the ACE2 protein. Furthermore, we selected the optimal conformation of the drug in the IFD results for molecular dynamics simulation. We found that bergamottin could exist in the cavity where the active site of ACE2 protein was located, and the protein binding of bergamottins and ACE2 was mainly water bridge and hydrophobic bond, suggesting that the binding stability was relatively low. However, PAP-1 could bind to the cavity of ACE2. The analysis of PAP-1 and ACE2 binding stability showed that the conformational change was small. Compared with bergamottin, PAP-1 has better binding capacity. Indirectly, PAP-1 and bergamottin can preferentially bind to ACE2.

Representative models to study virus–host interactions in the lungs are not available. These models should be based on human cells relevant to the disease in order to study the behavior of SARS-CoV-2 and possible pharmacological interventions. Alveolar type II cells express 83% ACE2, which is essential for SARS-CoV-2 cell entry for genes encoding host cell proteins [26]. In contrast, alveolar epithelial type II cells showed high expression of ACE2, suggesting that these cells can serve as reservoirs for viral invasion. In addition, gene ontology enrichment analysis showed that alveolar epithelial type II cells expressing ACE2 had a high level of multiple genes related to viral processes, including regulatory genes for viral processes, viral life cycles, viral assembly, and viral genomic replication, suggesting that ACE2 expressing AT-II cells promoted coronavirus replication in the lung [27]. In an animal model of SARS-CoV infection, the overexpression of human ACE2 can lead to the infection and replication of SARS-CoV-2, aggravating lung injury and enhancing the severity of the disease [28]. Importantly, this damage is attenuated by blocking the renin–angiotensin pathway and is dependent on ACE2 expression. High levels of ACE2 expression are associated with increased SARS-CoV-2 infection.

In a recent study, Yu [29] reported that receptor binding acted as the upstream scene of membrane fusion, and the inhibition on receptor binding or expression would lead to a decrease in membrane fusion. We further tested the effects of bergamottin and PAP-1 on receptor expression while it inhibited the cellular ACE2 expression in AT-II. We therefore caution against the use of bergamottin and PAP-1 in high dosages until its effects on SARS-CoV-2 spike activation are better understood.

Our results showed that ACE2 protein was expressed in AT-II cell lysates as a single 110 kDa band, which is consistent with reports in the literature [30]. Furthermore, 150 µM PAP-1 and 70 µM bergamottin significantly reduced ACE2 mRNA and proteins by 67%, 58% and 55%, 41%, respectively. Compared to bergamottin, PAP-1 induced the greatest inhibitory effect. This is consistent with reports that bergamottin inhibits SARS-CoV-2 infection in Caco-2 cells with decreased ACE2 expression [31].

In general, all findings support cytokine storm and severe fibrosis in the lungs of COVID-19 patients [32]. The main performance is AT-II cell hyperplasia with no evidence of hyaline membranes [33]. Bergamottin and PAP-1 inhibit the proliferation of AT-II cells in vitro, but they can damage the morphology of human alveolar type II epithelial cells. These results may mean that they affect the tissue repair process; further research in this area is required.

In conclusion, our results indicate that bergamottin and PAP-1 could be potential inhibitors of SARS-CoV-2, which could potentially be used in combination with other drugs to improve the efficacy of the antiviral effect. In addition, we provided evidence that bergamottin and PAP-1 are effective against SARS-CoV-2 at micromolar concentrations with a high selective concentration in alveolar type II cells. These results indicate that the psoralen compounds, PAP-1 and bergamottin binding to the ACE2 protein, could be further developed in the fight against COVID-19 infection during the current pandemic. However, the dose of compounds PAP-1 and bergamottin should be carefully considered. Further research is required to determine if these treatments can be used effectively against the coronavirus.

## 4. Materials and Methods

### 4.1. Cell Lines and Antibody Proteins

Human alveolar type II epithelial cells were purchased from the Chinese Academy of Sciences, Beijing, China. The antibody against ACE2 (ab108252) was purchased from American Abcam, Waltham, MA, USA. Rabbit anti-beta actin and goat anti-rabbit IgG (H + L)/HRP were purchased from Chinese Bioss Science Company, Beijng, China. The human ACE2/angiotensin-converting enzyme 2 protein (His Tag 10108-H08H) and SARS-CoV-2 Spike RBD-mFc recombinant protein (40592-V05H) were purchased from Chinese Sino Biological Science Company, Beijing, China. All other reagents of cell culture were purchased from Gibco, Waltham, MA, USA.

### 4.2. Inhibitors

PAP-1 and bergamottin, two small molecular compounds, were purchased from the MCE company, and their purities reached 99.52% and 99.80%, respectively.

### 4.3. Molecular Docking

The components of furanocoumarins were obtained from TCMSP (Traditional Chinese Medicine Systems Pharmacology Database and Analysis Platform, http://tcmspw.com/tcmsp.php, accessed on 15 February 2021) and YATCM (http://cadd.pharmacy.nankai.edu.cn/yatcm/home, accessed on 15 February 2021). The criteria for candidate compound were the following: (a) oral bioavailability (OB) ≥ 30%; (b) drug-likeness (DL) ≥ 0.18; and (c) lipid/water partition coefficient (AlogP) < 5. The crystallographic structure of the RBD Spike S protein (PDB code: 7DF4) [21] and ACE2 protein were obtained from the PDB bank. Psoralen groups of PAP-1 and bergamottin were optimally docked with ACE2 by AutoDock, Discovery Studio, and Sybyl docking software (SYBYL-X 2.1.1, Tripos, Jacksonville, FL, USA). The common features of protein preparation were adding hydrogen atoms and repairing missing atoms. Total residues were checked for cap termini and to obtain scoring function and Binding energy. Docking parameter files were written out according to the standard protocol [34,35].

### 4.4. Bio-Layer Interferometry Assay (BLI)

The Octet RED96e system (Molecular Device, ForteBIO, Fremont, CA, USA) is ideally suited for the characterization of protein–protein and protein–small molecule binding kinetics and binding affinity. The assay involved the following steps: (a) protein and bio-layer interferometry (BLI) sensor preparation. Binding affinity of PAP-1 and bergamottin for RBD spike S or ACE2 protein determined by biolayer interferometry assay using an Octet RED96 (ForteBio, Fremont, CA, USA). Ni-NTA biosensor tips (ForteBio, Fremont, CA, USA) were used for labeled proteins after dilution in a kinetic buffer (PBS, 0.1% bovine serum albumin, 0.02% Tween 20), the pH of buffer solution was adjusted to 7.4. The equilibrated Ni-NTA biosensors were loaded with RBD Spike S (100 nM) and ACE2 proteins (10 µg/mL); (b) BLI experimental process. Automated detection was performed using a RED96 instrument. All assays were performed by a standard protocol in 96-well black plates with a total volume of 200 µL/well at 30 °C. Baseline readings were obtained in buffer (120 s), associations in wells containing compound (180 s), and dissociation in buffer (180 s). The signals from the following buffer were detected over time. All the data were analyzed by Octet Data Analysis Software 10.0. The signals were analyzed by a double reference subtraction protocol to deduce non-specific and background signals and signal drifts caused by biosensor variability. Equilibrium dissociation constant (Kd) values were calculated from the ratio of K-off to K-on.

### 4.5. Molecular Dynamic Model Simulation of Interaction between PAP-1 and Bergamottin with Proteins

Root-Mean-Square Deviation, Root-Mean-Square Fluctuation and Contacts. The 3D structure of ACE2 protein was downloaded from the PDB database, and the bergamottin and PAP-1 structures were derived from the Pubchem database. In order to analyze and compare the effects of two drugs on the ACE2 active site, we first selected the active site for the binding of ACE2 and the spike protein of COVID-19 and used the Glide module of Schrodinger software to blind dock the two drugs with the ACE2 protein. Induced-fit docking was then performed using an IFD module that allowed rotation of residues near the protein-binding pocket. We selected the optimal conformation of the drug in the IFD results for molecular dynamics simulation. During the simulation, we evaluated the root-mean square deviation of protein and drug in the system, the root-mean-square structure and ligand contacts. For the 200 ns simulation, the time trajectory was kept at 10.0 with approximately 1000 frames. The NPT ensemble class was selected, with 300 k (temperature) and 1.013 bar (pressure). The average RMSD for bergamottin was found to be 1.191 and 0.621 for PAP-1. Fluctuations were noted within the acceptable range at a few points, but overall complex for both proteins remained stable during the MD simulation (200 ns), while the ligand remained intact during the simulation with few deviations.

### 4.6. Cell Culture and Proliferation Assay

Human alveolar type II epithelial (AT-II) cells (5 × 10^3^/well) were maintained in DMEM basic 1× (4.5 g/L D-glucose) supplemented with 10% (*v*/*v*) fetal bovine serum (FBS), 100 μg/mL streptomycin, and 100 U/mL penicillin at 37 °C with 5% CO_2_ in 96-cell plates for 24 h. The supernatant of the cells was discarded. Then AT-II cells were subjected to starved incubation without FBS DMEM for 8 h. Then cells were incubated with serial dilutions of bergamottin with PAP-1 for 24 h, 48 h, and 72 h. Then the detection reagent CCK-8 was added 10 µL/well and incubated for 2 h. The absorbance at 450 nm was measured by a microplate reader (Molecular Devices, Thermo Fisher, Waltham, MA, USA).

### 4.7. Western Blot Analysis

AT-II cells were washed with PBS three times. After being digested with pancreatic enzyme for 1 min, transferred to a 1.5 mL EP tube, and having used DMEM to stop digestion, they were centrifuged at 1000 rpm for 5 min. The supernatant was then discarded, and the cells precipitation was extracted using a RIPA Lysis buffer (Thermo Fisher Scientific, Waltham, MA, USA) containing a protease and phosphatase inhibitor cocktail to obtain the protein lysates. The lysate concentration was measured with the BCA Protein Assay Kit (Solarbio, Beijing, China). An amount of 20 μg lysate protein from each sample was separated via 8% SDS-PAGE at 100 V for 90 min and transferred onto PVDF membrane (Millipore, Billerica, MA, USA) at 80 V for 150 min. After being blocked with 5% (*v*/*v*) skimmed milk in 1× Tris-buffered saline-Tween-20 (TBST) for 1 h and washed three times for 30 min in 1 × TBST, membranes were incubated with primary antibodies at 4 °C overnight with ACE2 (1:1000) and β-actin (1:2000). Probed membranes were washed three times for 30 min in 1 × TBST and incubated with rabbit anti-beta actin and goat anti-rabbit IgG (H + L)/HRP secondary antibody (1:5000) (Bioss, Beijing, China) for 1 h at room temperature. Protein bands were visualized using the ECL chemiluminescence solution kit (Biosharp, Beijing, China) and band densities were scanned and calculated using Image Lab 5.2.1 Software (Bio-Rad, Hercules, CA, USA). Database search results were loaded onto Image J 1.8.0 (National Institutes of Health, Bethesda, MD, USA) for spectral counting, statistical analysis, and data visualization. The histograms corresponding to the blots were normalized to β-actin.

### 4.8. Quantitative Real-Time PCR (qPCR) Analysis

Total RNA was extracted from the AT-II cells. The cDNA was reverse transcribed using Thermo Scientific Revert Aid 1st Strand cDNA Synthesis Kit (00970909, Thermo Fisher, Waltham, MA, USA) following the manufacturer’s instructions. The RT-PCR for mRNAs was performed on a Step One™ Real-Time PCR System (Life technologies, Carlsbad, CA, USA) using QuantinNava™ SYBR^®^ Green PCR Super Mix (208054, Qiagen Biotechnology, Hilden, Germany). The reaction mix system was prepared as follows: SYBR Mix 10 µL, primer forward 1 µL, primer Reverse 1 µL, ddH_2_O_2_ 6 µL, and ACE2 cDNA 2 µL, with a total reaction volume of 20 µL. The reaction conditions were as follows: 95 °C for 2 min, 95 °C for 5 s, 60 °C for 20 s, with a total of 40 cycles. The relative expression levels of mRNA were normalized to those of GAPDH and evaluated using the 2^−ΔΔCT^ method. The primers used for RT-PCR are listed in Table 5.

## Figures and Tables

**Figure 1 ijms-23-12565-f001:**
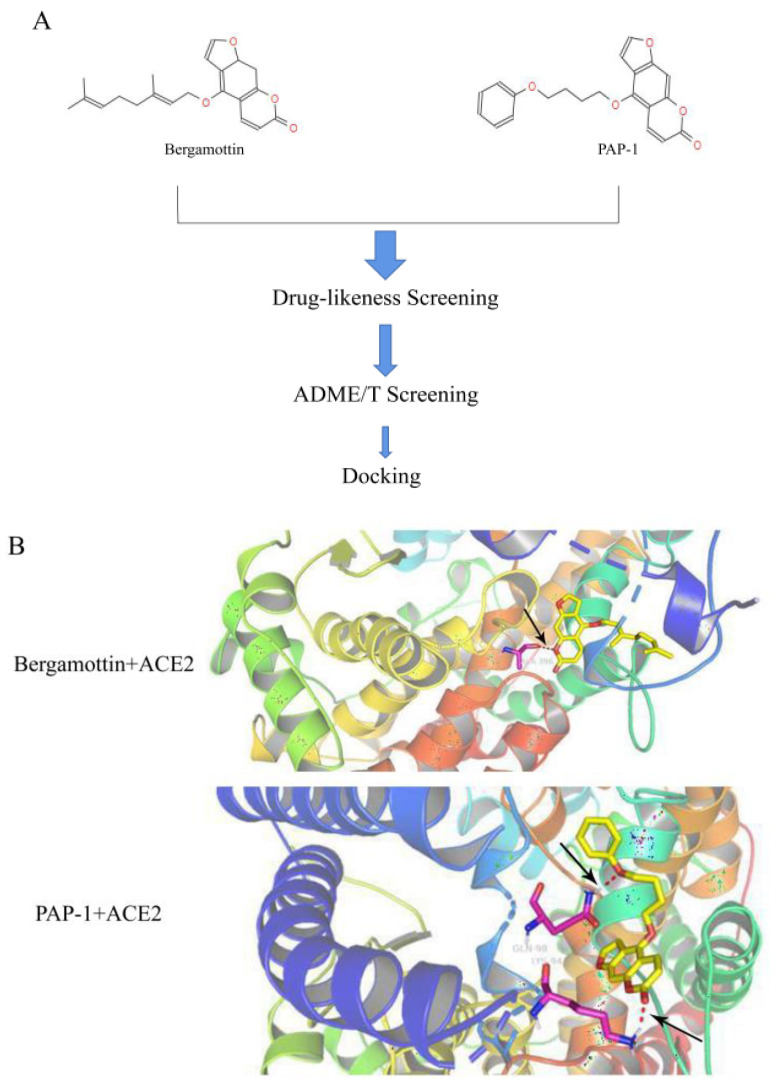
(**A**) Chemical structures of bergamottin and PAP-1. (**B**) Bergamottins and PAP-1 via utodocking tools to predict the binding of the analogue molecules with ACE2 docking pose.

**Figure 2 ijms-23-12565-f002:**
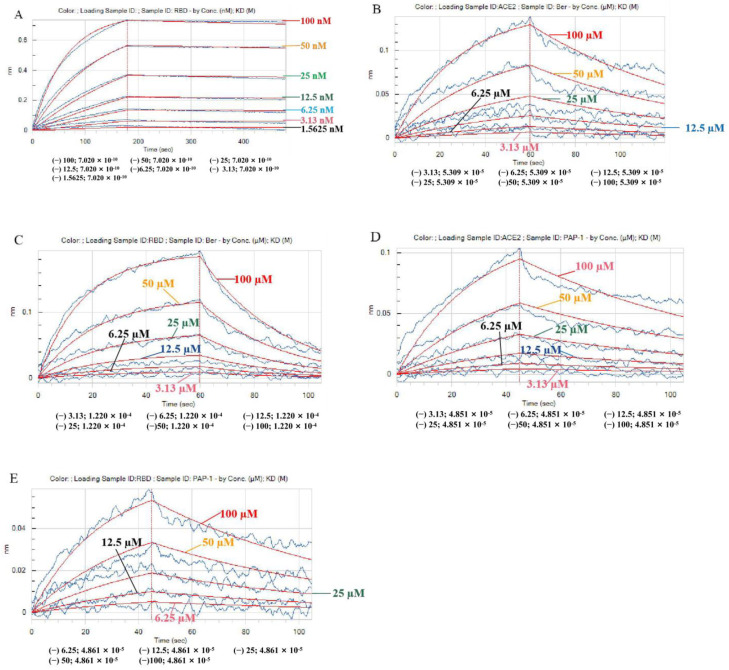
(**A**) The interaction between the RBD Spike S protein and recombinant ACE2 protein was analyzed by BLI assay; binding affinity using an Octet RED96e system. Seven different RBD Spike protein concentrations including 1.56, 3.13, 6.25, 12.5, 25, 50, and 100 nM were set. Response (y axis) was measured as a shift in nm in the interference pattern and was proportional to the number of molecules bound to the surface of the biosensor. Response was recorded and displayed on a sensorgram in real time. The data points are represented by blue lines and the fitted data are shown as red lines. (**B**) Interaction between bergamottin and recombinant ACE2 protein was analyzed by BLI assay. Six different bergamottin concentrations including 3.13, 6.25, 12.5, 25, 50, and 100 µM were set. (**C**) Binding affinity measurements of bergamottin and RBD Spike S protein using BLI. Representative sensorgrams obtained from the injection of different concentrations of bergamottin (3.13, 6.25, 12.5, 25, 50 and 100 μM). (**D**) Interaction between PAP-1 (3.13, 6.25, 12.5, 25, 50 and 100 µM) and recombinant ACE2 protein was analyzed by BLI assay. (**E**) Interaction between PAP-1 (6.25, 12.5, 25, 50 and 100 µM) and recombinant RBD Spike protein was analyzed by BLI assay.

**Figure 3 ijms-23-12565-f003:**
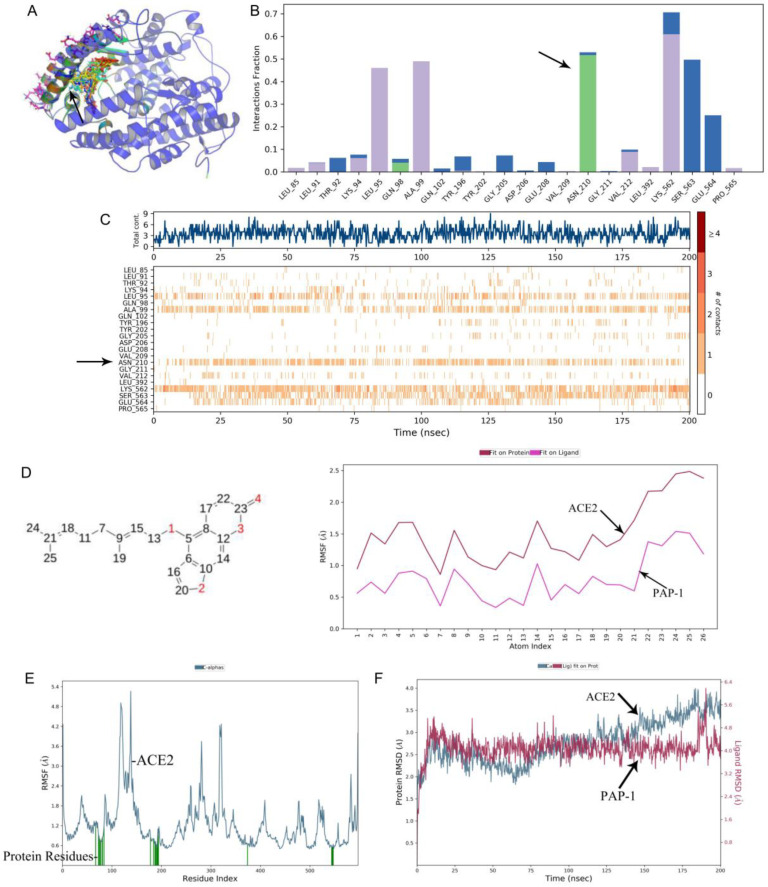
Molecular dynamic simulation analysis of PAP-1 and ACE2. (**A**) Molecular docking result mode. The black arrow shows the binding position of ACE2 protein with PAP-1. (**B**) Analysis of interaction between ACE2 protein residue and PAP-1. H-bonds (green part) play a significant role in ligand binding. The black arrow shows that the ACE2 protein has made more than one specific contact with the ASN210 residue in a hydrogen bond. (**C**) Contract number with protein residues. The black arrow shows ASN210 residues make more than one specific contact with the ligand, which are represented by a darker shade of orange, according to the scale to the right of the plot. (**D**) Drug action analysis of RMSF. The black arrow shows the internal atom fluctuations of the PAP-1 (**E**) ACE2 protein RMSF. The black arrow shows ACE2 protein residues that interact with the PAP-1 are marked with green-colored vertical bars. (**F**) The interaction of RMSD between ACE2 protein with PAP-1. The black arrow indicates that the observed ligand PAP-1 value is significantly less than the RMSD of protein, indicating that PAP-1 did not diffuse from its initial binding site.

**Figure 4 ijms-23-12565-f004:**
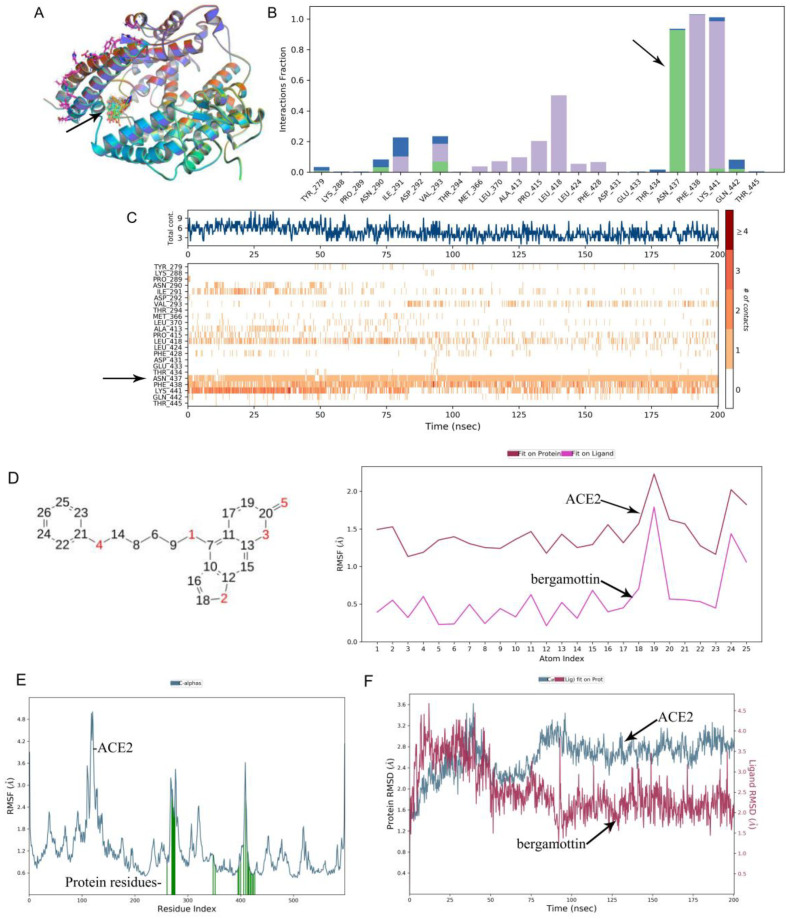
Molecular dynamic simulation analysis of bergamottin and ACE2. (**A**) Molecular docking result mode. The black arrow shows the binding position of ACE2 protein with bergamottin. (**B**) Analysis of interaction between ACE2 protein residue and bergamottin. H-bonds (green part) play a significant role in ligand binding. The black arrow shows that the ACE2 protein has made more than one specific contact with the ASN437 residue in a hydrogen bond. (**C**) Contract number with protein residues. The black arrow shows ASN437 residues make more than one specific contact with the ligand, which is represented by a darker shade of orange, according to the scale to the right of the plot. (**D**) Drug action analysis of RMSF. The black arrow shows the internal atom fluctuations of the bergamottin. (**E**) ACE2 protein RMSF. The black arrow shows ACE2 protein residues that interact with the bergamottin are marked with green-colored vertical bars. (**F**) The interaction of RMSD between ACE2 protein with bergamottin. The black arrow indicates that the observed ligand PAP-1 value is significantly less than the RMSD of protein, indicating that bergamottin did not diffuse from its initial binding site.

**Figure 5 ijms-23-12565-f005:**
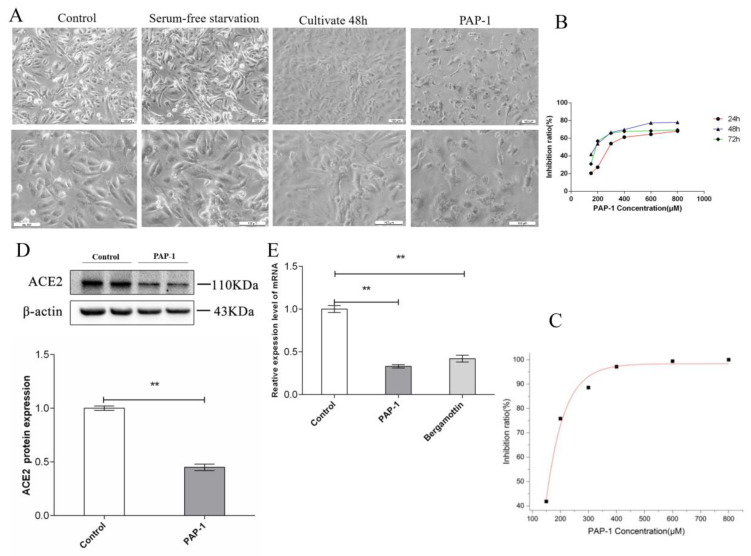
Effects of PAP-1 viability of the AT-II cells and cell morphology. (**A**) AT-II were treated with the indicated concentrations of PAP-1 for 24 h for cell morphology (×100 and ×200). (**B**) Effects of bergamottin on viability of AT-II cells for 24, 48,and 72 h by CCK-8 kit, comparison of cell viability at 48 h of PAP-1, the indicated concentrations of PAP-1 for 48 h. (**C**) After 48 h treatment of AT-II cells with PAP-1 at 150, 200, 300, 400, and 800 µM, cell viability (by CCK-8 assay, *n* = 5) of both cell types was inhibited in a concentration-dependent manner in accordance with the Hill equation: y = ax^b^/(c^b^ + x^b^). The IC50 for AT-II cells were calculated as 150 µM. An amount of 150 µM PAP-1 suppressed cell viability in AT-II cells, after which the viabilities stabilized. Accordingly, 150 µM was chosen as the PAP-1 dose for subsequent experiments. (**D**) Western blot detection of expression of ACE2 proteins in AT II cells. Normalized to β-actin. ** *p* < 0.01, PAP-1 vs. Control, ACE2 protein after administration of PAP-1 were signifificantly lower than the control group. (**E**) mRNA expression levels of ACE2 relative to AT-II cells with PAP-1. The data in the figure was measured in the following manner: Unpaired data in compliance with normal distribution and homogeneity between two groups and were compared using an unpaired *t* test. The experiment was independently repeated three times.

**Figure 6 ijms-23-12565-f006:**
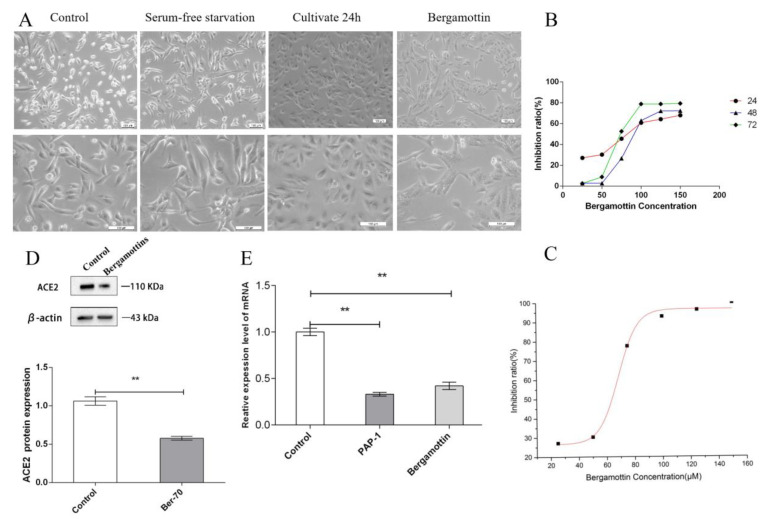
Effects of bergamottin viability of AT-II cells and cell morphology. (**A**) AT-II cells were treated with the indicated concentrations of bergamottin for 24 h with cell count (×100 and ×200). (**B**) Effects of bergamottin on viability of AT-II cells for 24, 48, and 72 h by CCK-8 kit, comparison of cell viability at 24 h of bergamottin with different concentrations. (**C**) After 24 h treatment of AT-II cells with bergamottin at 25, 50, 75, 100, and 150 µM, cell viability (by CCK-8 assay, *n* = 5) of both cell types was inhibited in a concentration-dependent manner in accordance with the Hill equation: y = ax^b^/(c^b^ + x^b^). Accordingly, 70 µM was chosen as the bergamottin dose for subsequent experiments. (**D**) Western blot detection of expression of ACE2 proteins in AT-II cells. Normalized to β-actin. ** *p* < 0.01 bergamottin vs. Control, ACE2 protein after administration of bergamottin were signifificantly lower than the control group, (**E**) mRNA expression levels of ACE2 relative to AT-II cells with bergamottin. The data in the figure was measured in the following manner: Unpaired data in compliance with normal distribution and homogeneity between two groups were compared using an unpaired *t* test. The experiment was independently repeated three times.

**Figure 7 ijms-23-12565-f007:**
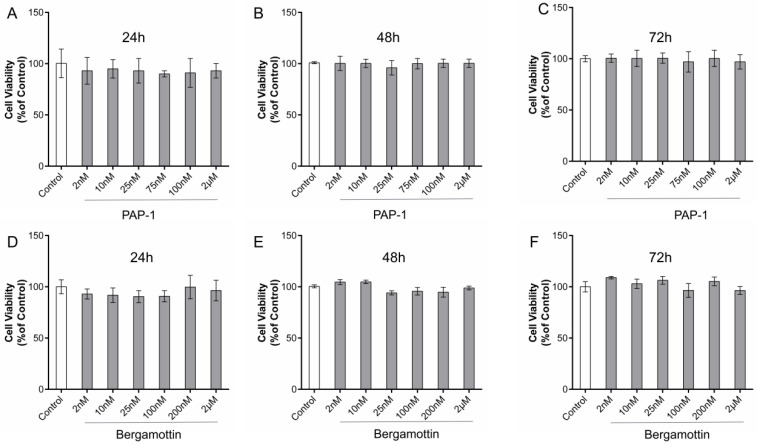
Effects of PAP-1 and bergamottin on viability of AT-II cells (CCK-8 assay). (**A**–**C**) Effects on viability of AT-II cells stimulated with PAP-1 for 24, 48, and 72 h. (**D**–**F**) Effects on viability of AT-II cells stimulated with bergamottin for 24, 48, and 72 h. Data are presented as mean ± SD, *n* = 3 (three separate experiments).

**Table 1 ijms-23-12565-t001:** Bergamottin and PAP-1 docking parameters.

ID Chemical Formula	Name	TCMSP			Auto Dock	Discovery Studio		SYBYL	
MOL001951	Bergamottin	OB (%)	DL (%)	AlogP	Binding energy (kcal/mol)	LibDockScore	Binding Energy	Total-score	C-score
C_21_H_22_O_4_									
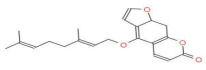		41.73	0.42	5.48	−8.18	131.25	4982.11	6.79	4
ZINC13829441	PAP-1	-	-	4.78	−6.54	135.24	1242.96	5.59	5
C_21_H_18_O_5_									
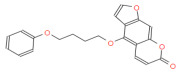									

**Table 2 ijms-23-12565-t002:** The interaction between compounds and ACE2 proteins.

Compounds	KD(M)	Kon (1/Ms)	Kdis (I/s)	Full R^2^
PAP-1	4.85 × 10^−5^	2.40 × 10^2^	1.17 × 10^−2^	0.96
Bergamottin	5.31 × 10^−5^	2.38 × 10^2^	1.26 × 10^−2^	0.96

**Table 3 ijms-23-12565-t003:** The interaction between compounds and RBD Spike S proteins.

Compounds	KD(M)	Kon (1/Ms)	Kdis (I/s)	Full R^2^
PAP-1	4.86 × 10^−5^	2.59 × 10^2^	1.26 × 10^−2^	0.96
Bergamottin	1.22 × 10^−4^	2.73 × 10^2^	3.33 × 10^−2^	0.99

**Table 4 ijms-23-12565-t004:** Docking results of Bergamottin and PAP-1 with ACE2.

Compounds	Glide Score (kcal/mol)	IFD-Glide-Score (kcal/mol)
PAP-1	−4.332	−8.439
Bergamottin	−4.788	−7.289

**Table 5 ijms-23-12565-t005:** Gene primer set information.

Gene ID	Forward (5′ to 3′)	Reverse (3′ to 5′)
GAPDH	CGGATTTGGTCGTATTGGG	TCTCGCTCCTGGAAGATGG
ACE2	CATTGGAGCAAGTGTTGGATCTT	GAGCTAATGCATGCCCATTCTCA

## Data Availability

All relevant data and materials are available from the authors upon reasonable request.

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
