# Peer review of "Bergamottin and PAP-1 Induced ACE2 Degradation to Alleviate Infection of SARS-CoV-2"

_ijms, 2022, doi:10.3390/ijms232012565_

Round 1

Reviewer 1 Report

The paper describes the binding ability of Bergamottin and PAP-1 compounds to ACEII proteins that was calculated by several docking software and the Octet system. These data showed a high affinity of these compounds to ACES II. However, in vivo analysis of these compounds require high concentration to inhibit cell growth. There is no evidence that the binding to ACEII of the compounds affect this cell growth. The mechanism of the downregulation of ACEII is also unclear. The authors should provide some comments in the text on the relationship between docking study and cell assay, especially ACEII downregulation mechanisms.

Author Response

Thank you  for your comments,Please see the attachment.

Reviewer 2 Report

In the current article, the authors showed the possibility of including the Psoralen compounds PAP-1 and Bergamottin as active compounds for the prevention and treatment of coronavirus infection (COVID-19) using molecular dynamic simulation analysis and cell experiments. They confirm that human alveolar type-Ⅱ epithelial cells showed high expression of ACE2 proteins across all human alveoli and report as a preliminary prediction that Bergamottin and PAP-1 had a high binding ability with ACE2 protein. As a result of the innate anti-virus immune response, a dynamic reaction of the human body is observed expressed in a virus-mediated cytokine storm, which leads to various pathologies changes in cells and tissues, organ damage, and even death. It is established that oxidative stress and а generation of high levels of free radicals, can lead to damage in cells, tissues, and organs and play a major role in the pathology of COVID-19. I would recommend that the authors consider in a future article the anti-oxidative effects of Bergamottin and PAP-1 on COVID-19-induced oxidative stress. 

Author Response

(The authors gave the same response as above.)

Reviewer 3 Report

The authors present a study to determine the anti-viral effects of Bergamottin and PAP-1, compounds that have an affinity for the ACE2 protein to which the spike protein of SARS-Cov2 binds to enter cells.  The authors investigated the ability of these two compounds to bind to ACE2, the binding energy and the effect of these two psoralens on the viability of human alveolar type II epithelial cells.  

1. The authors investigated psoralen compounds and found that Bergamottin and PAP- 1 were the preferred molecules based on the binding energy, drug-likeness properties and binding energies to ACE2.  Authors in Fig 1D, it would be helpful if by color or arrows you indicate where these compounds are binding.  The current figure makes this difficult to determine.  In Fig 2, all the concentrations are in blue lines, can you use different colors to indicate the various concentrations?  Difficult to associate concentrations with each line.

2.  Used a molecular dynamic model simulation to verify that Bergamottin and PAP-1 binds to ACE2 protein.  Again, need to use arrows or some other indicator to show where the psoralens are binding to the ACE2.  The experiments in number 1 and 2 do establish the ability of these two compounds to bind to the ACE receptor.

3.  Interaction with AT-II cells in vitro.  These experiments used an alveolar cell line that expresses the ACE2 receptor.  The data indicates a dosage dependent response to the toxicity of each compound.  Authors viewing the bar graphs in Figure 5 A and B there does not appear to be much variation on cell viability compared to the control.  The error bars in each bar appear to overlap indicating no significant difference in the percent viability irrespective of the concentration of the Bergamottin and PAP-1.  Perhaps this reviewer is misinterpreting your data?  Your data definitely shows that the mRNA production of ACE2 and quantity of protein expression on Western blots was reduced by each compound?  How do you explain what appears to be a relative minimal loss of viability with the results of lower mRNA and protein production?

Overall, this paper does present interesting data on two compounds that have a definitely interfere with the binding of SARS-CoV 2 to the ACE2 receptor.  If these compounds can be utilized at concentrations that do not have a toxic effects on the cells, as indicated in vitro, they may have potential as antivirals versus SARS-CoV 2.  The advantage of this drug therapy is that since all known SARS CoV 2 viruses use the ACE2 receptor to initiate infection, their interaction may be effective versus all variants of the virus.  

Finally,  authors the title is rather lengthy and needs to be changed.  One suggestion is Bergamottin and PAP-1 bind to the ACE2 receptor for SARS CoV 2 Interfering with Viral Replication or something similar.  No need to include the cytotoxicity of the cells in the title in the opinion of this reviewer.  

Author Response

(The authors gave the same response as above.)

Reviewer 4 Report

The manuscript describes both computational and biological studies of PAP-1 and bergamottin selected by virtual screening as ACE2 inhibitors. In my opinion, the content of the paper is not adequate to the standard level of the Int. J. Mol. Sci.

The paper is very hard to read and difficult to understand. The English grammar and sentence construction need a deep revision.

A suggestion to improve the merit of the work: in the proliferation assays on AT-II cells,  the tested compounds showed activity at concentrations in the high micromolar range. I wonder what the value of these results is. It could be of help evaluation of a reference compound.

I am not able to evaluate the correctness and value of docking studies because my expertise doesn’t cover the computational study area.

Author Response

Thank you for your comments,please see the attachment.

Round 2

Reviewer 3 Report

The authors have addressed most of the questions posed by this reviewer.  The suggested title change is accepted, still would like to see different colored lines for the graphs with the various concentration of the compounds.  When lines are all the same color, difficult to determine which concentration correspond to which lines (perhaps simply put the concentration number next to the lines?)  I understand the issue with how the reader presents the data, but you can label each line.

The grammar is markedly improved but may still need s review.  

Reviewer 4 Report

The revised manuscript has been improved, it  is clearer and easier to read. Thus, it can be published
